# Study on the Material Basis of Neuroprotection of *Myrica rubra* Bark

**DOI:** 10.3390/molecules24162993

**Published:** 2019-08-18

**Authors:** Shengnan Shen, Mengjun Zhao, Chenchen Li, Qi Chang, Xinmin Liu, Yonghong Liao, Ruile Pan

**Affiliations:** Institute of Medicinal Plant Development, Chinese Academy of Medical Science, Peking Union Medical College, Beijing 100193, China

**Keywords:** *Myrica rubra*, UPLC-Q-TOF-MS, glutamate, neuroprotection, myricitrin, myricanol 11-sulfate

## Abstract

**Background**: Increasing attention has been given to the search for neuroprotective ingredients from natural plants. *Myrica rubra* bark (MRB) has been used in traditional oriental medicine for over thousand years and has potential neuroprotection. **Methods and Results**: Ultra-performance liquid chromatography quadrupole time-of-flight mass spectrometry (UPLC-Q-TOF-MS) was used to identify the compounds in MRB extract, and the MTT assay was performed to evaluate the neuroprotection of six major compounds from MRB against glutamate-induced damage in PC12 cells. The result displayed nineteen compounds were identified, and myricitrin and myricanol 11-sulfate were shown to have neuroprotection, which prevented cell apoptosis through alleviating oxidative stress by reducing the levels of reactive oxygen species and methane dicarboxylic aldehyde, as well as by enhancing the activities of superoxide dismutase. **Conclusions**: Several active compounds from MRB may offer neuroprotection and have the potential for the development of new drugs against central nervous system diseases.

## 1. Introduction

In the mammalian central nervous system (CNS), glutamate is the main excitatory neurotransmitter, and is involved in many aspects of normal brain function, including cognition, learning, memory, and in synaptogenesis [1]. However, when neurological insults occur, such as stroke (anoxia/ischemia), depression, epilepsy and neurodegenerative disorders (Alzheimer’s disease and Parkinson’s disease), excessive synaptic glutamatergic transmission will occur. There are two different mechanisms of glumate toxicity: One is excitotoxicity leading to superoxide production, which affects Ca^2+^-mediated NO production and causes mitochondrial dysfunction, the other is oxidative toxicity of glutamate which resulted in the reduction of the cystine (Cys) glutamate antiporter. Reduced Cys affects the ability of glutathione (GSH) to scavenge free radicals, leading to cell death and neuronal damage [2,3,4]. Therefore, a neuroprotective strategy against glutamate-induced injury has become an important target for diseases of the central nervous system. Moreover, due to the side effects of synthetic compounds, more and more attention has been paid to searching for natural plants.

*Myrica rubra* (Family Myricaceae) is an important plant, which has been cultivated for more than two thousand years in southern China [5]. *Myrica rubra* bark (MRB) has been used for the treatment of duodenal ulcers, stomachache and swelling, and bruises [6]. Chemical studies have reported that MRB contains flavonoids, diarylheptanoids, and triterpenes [7,8,9,10,11]. Pharmacological studies on MRB showed that its methanolic extract had protective effects on CCl_4_- and α-naphthylisothiocyanate-induced liver injury [12], and the 50% ethanolic extract inhibited melanin biosynthesis [13]. The diarylheptanoids and phenolic compounds isolated from MRB were found to inhibit induction of NO synthase and overproduction of reactive oxygen species (ROS) [14,15], which contribute to many degenerative nerve diseases. The main compound from MRB, myricitrin (myricetin-3-O-a-rhamnoside), was shown to possess a variety of potential health benefits, such as anti-oxidative [16], anti-inflammatory [17,18], and anti-nociceptive effects [18,19,20], and cardiovascular protection [21,22]. In our study, we found that MRB extract displayed good protection against glutamate-induced damage in PC12 cells (data not shown). Although MRB has drawn a lot of attention, there are few reports on the composition using Ultra-performance liquid chromatography quadrupole time-of-flight mass spectrometry (UPLC-Q-TOF-MS), and its neuroprotective constituents are still not clear.

In the present study, UPLC-Q-TOF-MS was used to identify the chemical constituents from MRB extract. Additionally, to discover potentially neuroprotective agents, six major compounds from MRB extract were evaluated the neruoprotective effects against glutamate-induced damage in PC12 cells by methyl thiazolyl tetrazolium (MTT) assay.

## 2. Results and Discussion

### 2.1. Identification of the Chemical Constituents of MRB by UPLC-PDA and UPLC-Q-TOF-MS Analysis

UPLC-Q-TOF-MS, which is an efficient and sensitive method, has been used extensively in the qualitative and quantitative analysis of most plant chemical constituents. In our study, a UPLC-Q-TOF-MS method was used to qualitatively analyze the chemical constituents from MRB, and a total of 19 compounds were tentatively identified. Typical UPLC-MS profiles in the positive-ion mode of MRB are shown in Figure 1. The peaks were classified into four groups, namely phenolic acids (Rt from 0 to 2.5 min), flavonoids (Rt from 2.5 to 7 min), diarylheptanoids (Rt from 7 to 15 min) and triterpenoids (Rt from 24 to 30 min). These were identified through the precise molecular mass, MS/MS data, UV-visible spectral characteristics, and by comparing the retention times with those of the available standards. The 19 tentatively identified compounds are shown in Table 1. They include one phenolic acid (gallic acid), eight flavonoids (rutin, myricetin hexoside, quercetin hexoside, myricitrin, quercetin deoxyhexoside, myricetin, quercetin, kaempferol), six diarylheptanoids (three myricanol hexosides, myricanol 11-sulfate, myricanol, myricanone) and four triterpenoids (uosolic acid, myricadoil, uvaol, tarxerol).

#### 2.1.1. Phenolic Acid (Peak 1)

One phenolic acid was identified from the extract of MRB, as indicated by peak 1, with a retention time of 1.5 min. The UV λmax of peak 1 presented at 219.3 nm and 271.1 nm. ESI-MS spectra showed the molecular ion [M + H]^+^ at *m/z* 171.1212 and a fragment ion at *m/z* 127.0708, which corresponds to the loss of a carboxyl residue (mass unit 45) from the molecular ion. Peak 1 was unambiguously identified as gallic acid, which was also confirmed by comparing it with the authentic standards.

#### 2.1.2. Flavonoids (Peaks 2–9)

Several well-resolved flavonoid chromatographic peaks were observed from Figure 2. Under the experimental conditions mentioned earlier, eight flavonoids (peaks 2–9) were provisionally identified. It is well known that flavonoids exhibit two major absorption peaks at 240–400 nm (band II) and 300–380 nm (band I). Band II (240–400 nm) is considered to be associated with absorption of the A-ring benzoyl system, and Band II (300–380 nm) is considered to be associated with the absorption of the B-ring cinnamoyl system, such as the compound myricitrin (peak 5) with two absorption maxima at 260.8 and 337.9 nm (Figure 3). Peaks 3, 5 and 7 had molecular ions [M + H]^+^ at *m/z* 481.3438, 465.0664 and 319.0455, respectively. All three peaks also had a fragment ion at *m/z* 319.0451, which is the typical mass of myricetin aglycone in the positive mode (Figure 4), indicating that these three compounds were myricetin and its derivatives. According to the literature and chemical standard confirmation, peak 7 was identified as myricetin. It has been reported that sugars bound to the aglycons are hexoses with a mass unit of 162, and deoxyhexoses with a mass unit of 146. Peak 5 had a retention time of 5.4 min, and the fragment ion at *m*/*z* 319.0455 corresponded to the loss of a mass unit of 146 (the deoxyhexoses fragment). By comparing it with the authentic standard, peak 5 was identified as myricitrin, which was reported as one of the main flavonols in MRB, as shown in Figure 2. The ion products of MS/MS of peak 3 showed the same fragmentation pattern as peak 5, where the ion at *m/z* 319.04 [M + H − 162]^+^ resulted from the cleavage of a hexoside residue, which indicated that peak 3 was myricetin hexoside. Peaks 2, 4, 6, and 8 had a fragment ion at *m/z* 303.05, indicating that these compounds were quercetin and its derivatives. The ESI-MS spectra of peaks 4 and 6 indicated that their structure contained a hexosyl residue (465.0664 − 303.0526 = 162.0128) and a deoxyhexosyl residue (449.1089 − 303.0507 = 146.0582), respectively. By using the chemical standards, peak 2 was identified as rutin and peak 8 was identified as quercetin.

#### 2.1.3. Diarylheptanoids (Peaks 10–15)

As can be seen from Figure 5, the diarylheptanoids mass metabolites could be separated well in 7–15 min, and for peaks 10–15, six diarylheptanoids were provisionally identified. In our previous phytochemical studies, several diarylheptanoids, including myricanol 11-sulfate (peak 13), myricanol (peak 14), and myricanone (peak 15), were isolated from the MRB extract. By comparing with the standard, the UV–visible spectral characteristics of these compounds showed a λmax at around 231.5, 259.0 and 296.5 nm, respectively (Figure 6). LC-ESI-QTOF-MS analysis was carried out for further structure identification. Here, an ion at the retention time of 12.5 min (peak 14) is taken as an example to illustrate the identification process. The base peak of its [M + H]^+^ at *m/z* 359.1870 is indicative of the molecular formula C_21_H_26_O_5_. Additionally, the neutral loss of 18 Da showed a fragment ion at *m/z* 341.1761 was attributed to the characteristic ion [M + H − H_2_O]^+^ fragments (Figure 7). Peak 13 gave a molecular ion [M + H]^+^ at *m/z* 439.1414. It also showed the same fragment ion at *m/z* 341.1750, corresponding to loss of a H_2_O residue. Peak 15 gave a molecular ion [M + H]^+^ at *m/z* 357.1840 and showed a fragment ion at *m/z* 339.1603, corresponding to loss of a H_2_O residue. Peaks 10, 11 and 12 showed the same molecular ion [M + H]^+^ at *m/z* 521.2381. Among the ion products of MS/MS, the same fragmentation pattern was seen at *m/z* 359.1859 [M + H − 162]^+^ and 341.1768 [M + H − 162 − H_2_O]^+^, resulting from the cleavage of a hexoside residue from myricanol aglycone.

#### 2.1.4. Triterpenoids (Peaks 16–19)

As can be seen from Figure 8, the triterpenoids mass metabolites could be separated well in 24–30 min and, among them, peaks 16–19 were identified. According to the literature and chemical standard obtained in our previous phytochemical studies confirmation, peaks 16, 17, 18 and 19 were identified as uosolic acid, myricadoil, uvaol and tarxerol, respectively [11]. The composition of triterpene is complex and structural isomers may exist. Most triterpenoids in the genus Myrica belong to pentacyclic triterpenoids. For further structural analysis, nuclear magnetic resonance spectroscopy is still needed as the ESI-MS spectral data and UV–Vis spectral data were not enough.

### 2.2. Neuroprotection of Six Major Compounds from Glutamate-Induced Damage in PC12 Cells by MTT Assay

#### 2.2.1. Cell Toxicity Induced by Glutamate

According to the results of the MTT assay shown in Figure 9, the viability was gradually reduced when PC12 cells (a cell line derived from rat adrenal medulla pheochromocytoma) were exposed to glutamate with increasing concentration and time. These results show that the best concentration and time of glutamate exposure was 15 mM for 24 h, which gave cell viabilities of about 50% compared to the control cells.

#### 2.2.2. Effect of Single Compound on PC12 Cells under Different Concentrations

The effect of six compounds (myricitrin, quercetin-3-rhamnoside, quercetin, myricanol 11-sulfate, myricanol, myricanone) on PC12 cells under the concentrations of 1.25–10 μM was examined at 8, 12, 24, and 48 h. No significant changes were observed on the viability of cells treated with the three individual flavonoids (myricitrin, quercetin-3-rhamnoside, quercetin) at a concentration of 1.25−10 μM for 24 h, which was chosen for the next experiments. The three diarylheptanoids (myricanol 11-sulfate, myricanol, myricanone) at 10 μM showed slight cytotoxicity. According to this, we chose 5 μM as the maximum concentration of the individual diarylheptanoids for subsequent experiments (data not presented).

#### 2.2.3. Effect of Six Compounds on Glutamate-Induced Damage in PC12 Cells

As we can see from Figure 10, among the three flavonoids, myricitrin showed the best neuroprotective activity at the concentration-dependent range of 2.5–10 μM. When the cells were treated with 10 μM myricitrin for 24 h after exposure to glutamate, the cell viability reached 75.15 ± 3.23% compared to the glutamate-induced damage group. Among the three diarylheptanoids, glutamate-induced cytotoxicity could be inhibited by myricanol 11-sulfate at a concentration- dependent range of 1.25–5 μM. The other two compounds did not show protection. The cells treated with myricanol 11-sulfate at the concentration of 5 μM displayed the best neuroprotective activity, with a cell viability of 72.05 ± 2.09%. Therefore, myricitrin at the concentration of 10 μM and myricanol 11-sulfate at 5 μM were selected to do the subsequent experiments.

### 2.3. Effects of Myricitrin and Myricanol 11-Sulfate on Glutamate-Induced Apoptosis in PC12 Cells by Flow Cytometric Detection

The neuroprotective effect of myricitrin and myricanol 11-sulfate on glutamate-induced apoptosis in PC12 cells was determined via AnnexinV/PI double staining using flow cytometry detection. The Annexin V−/PI− population was regarded as normal cells, while the Annexin V+/PI− cells were taken as a measure of early apoptosis, and the Annexin V+/PI+ cells as necrosis/late apoptosis. As shown in Figure 11, the control group had 99.93% intact living cells and 0.07% of cells in early apoptosis. An increase in apoptotic cells was observed in the glutamate-treated group. When the PC12 cells were treated with myricanol 11-sulfate (5 μM) and myricetrin (10 μM), respectively, the cell apoptotic rate decreased (Figure 11C,D), which indicated that myricanol 11-sulfate or myricetrin inhibited glutamate-induced PC12 cell apoptosis.

### 2.4. Effect of Myricitrin and Myricanol 11-Sulfate on ROS, MDA and SOD Levels on Glutamate-Induced Oxidative Stress in PC12 Cells

As shown in Table 2, when PC12 cells were exposed to 15 mM glutamate for 24 h, the intracellular ROS and MDA contents were markedly increased to 122 ± 6%, (2.64 ± 0.3% compared with the control value) and 100 ± 1% (0.76 ± 0.1%), respectively. The activity of SOD was markedly decreased from 32 ± 4 units/mg prot (the control value) to 18 ± 1 units/mg prot with 10 μM myricetrin and 5 μM myricanol 11-sulfate, respectively, in the glutamate-induced damaged cells. At the same time, intracellular MDA and ROS contents were significantly reduced, and the SOD activity was increased compared with the glutamate group. Based on these results, the neuroprotection of myricanol 11-sulfate and myricetrin is likely to be due to reducing oxidative stress.

### 2.5. MRB Is a Potential Natural Source of Neuroprotection

Rat pheochromocytoma (PC12) cells, a cell line that has an embryonic origin from the neural crest, has been widely used to screen the neuroprotective constituents in vitro. In our preliminary study, it was found that MRB extract has neuroprotection against glutamate-induced cell damage in PC12 cells by the MTT test. To discover the potentially neuroprotective agents from MRB extract, the neuroprotection of six major compounds, which were isolated from MRB extract, were evaluated. Our results showed that myricitrin and myricanol 11-sulfate possessed marked neuroprotection against glutamate-induced damage in PC12 cells.

A previous study reported that myricitrin has many potential benefits, such as anti-oxidative [16], anti-inflammatory [17,18], and anti-nociceptive effects [18,19,20], as well as cardiovascular protection [21,22]. Myricitrin is also used as a flavor modifier in dairy products and beverages in Japan, and is listed as “generally recognized as safe” (GRAS) by the U.S. Flavor and Extract Manufacturer Association (FEMA) [23]. Specifications related to the identity and purity of myricitrin for use as a flavoring agent have been established by the Joint FAO/WHO Expert Committee on Food Additives (JECFA) [23,24]. Our present study revealed that myricetrin possessed neuroprotection against glutamate-induced apoptosis in PC12 cells. The result provided strong support for further studies exploring the potential neuroprotective effects of myricetrin in CNS diseases.

Diarylheptanoids are another major constituent of MRB. There are accumulating reports on their anti-inflammatory [25], antioxidant [26], antitumor [27], estrogenic and hepatoprotective effects [28]. This study is the first demonstration of the neuroprotective effects of myricanol 11-sulfate, a substituent group of diarylheptanoids. Of course, the structure–activity relationships of diarylheptanoids still need to be elaborated.

Additionally, we also showed that both myricanol 11-sulfate and myricetrin display neuroprotective activity through inhibiting oxidative apoptosis. The production of ROS markedly increased in glutamate treated PC12 cells. Treatment with myricanol 11-sulfate and myricetrin could largely alleviate the oxidative stress induced by glutamate in PC12 cells through reducing intracellular MDA and ROS content, respectively, as well as increasing the SOD activity.

In this study, both myriceirin and myricanol 11-sulfate showed neuroprotective effects in PC12 cell lines. Like many molecules (such as flavonoids) [29], myriceirin and myricanol 11-sulfate may have issues when crossing the blood–brain barrier (BBB). In future work, we will consider co-administering these compounds with α-tocopherol, which has been shown to promote transport of flavonoids across the BBB [30]. In addition, to enhance their penetration ability, these compounds will be coated with nanoparticles, such as glyceryl monooleate coated with various surfactants [31].

## 3. Materials and Methods

### 3.1. Drugs and Chemicals

Formic acid and acetonitrile (HPLC-grade) were obtained from Fisher Scientific (Pittsburgh, PA, USA). Ultrapure water (18.2 MΩ·cm, 25 °C) was obtained from a Milli-Q water purification system (Millipore, Molsheim, France). Dulbecco’s modified Eagle medium (DMEM), heat-inactivated horse serum, fetal bovine serum, penicillin and streptomycin were purchased from Gibco (Grand Island, NY, USA). Dimethylsulfoxide (DMSO), phosphate buffer (PBS), glutamate and MTT were purchased from Sigma-Aldrich (St. Louis, MO, USA). Assay kits for determination of SOD, Intracellular ROS and MDA were purchased from Nanjing Jiancheng Biotechnology Institute (Nanjing, China). FITC Annexin V/Dead Cell Apoptosis Kit with FITC Annexin V and PI were purchased from Zoman Bio Co., Ltd. (Beijing, China). All other chemicals were of analytical grade and all solutions for UPLC were filtered through 0.22 μm nylon membranes.

### 3.2. Materials

MRB was collected from Yuyao country, Zhejiang province, China in July 2012. The plant materials were authenticated by Professor Bengang Zhang of IMPLAD, Chinese Academy of Medical Sciences and Peking Union Medical College. The voucher specimen (2012061205) is deposited in the herbarium of IMPLAD. MRB were air dried in the dark at room temperature. Six major compounds; myricitrin (1), quercetin-3-rhamnoside (2), quercetin (3), myricanol 11-sulfate (4), myricanol (5), and myricanone (6) were isolated by our research group in a previous study. The purity of each compound is over 95% as confirmed by HPLC analysis.

### 3.3. Sample Preparations

The powdered MRB (about 100 g) was extracted with 250 mL methanol by ultrasound twice (each time for 30 min). The filtrates were combined and concentrated under vacuum. The yields of MRB were about 14.6 g.

The pure compounds (each about 1 mg), and MRB extract (about 10 mg) were weighed accurately, dissolved in 10 mL methanol solution (*v*/*v*), and then centrifuged at 13,000 *g* for 15 min, respectively. The supernatant was filtered through a 0.22 μm membrane filter before injection into the UPLC-MS system. All solutions were stored in the refrigerator at 4 °C and returned to room temperature before analysis.

### 3.4. UPLC-DAD and UPLC-ESI-Q-TOF-MS Analysis

A Waters Acquity TM Ultra Performance LC system (Waters Corporation, Milford, MA, USA) equipped with a photo diode array detector (DAD, 190–400 nm) was used for analysis, and the system was controlled by the Mass Lynx V4.1 software (Waters Co., Milford, MA, USA). Separations were performed using a Waters Acquity UPLC BEH C18 column (2.1 mm × 100 mm, 1.7 μm) at 40 °C, whereas the samples were maintained at 4 °C during the analysis. The mobile phase was composed of acetonitrile (A) and water (B), each containing 0.1% formic acid. A line gradient program was carried out as follows: 5–100% A at 0–30 min; 100% A at 30–35 min for column washing. The flow rate was 0.40 mL/min and the injection volume of the test sample was 1 μL. The mass spectrometric data were collected using a Q-TOF analyzer in a SYNAPT HDMS system (Waters Corporation, Milford, MA, USA) in a positive ion setting. The cone voltage was 30 V, the capillary voltage was 3000 V, the cone gas rate was 50 L/h, the desolvation gas rate was 800 L/h, the temperature was 420 °C, and the source temperature was 120 °C. The data acquisition rate was 0.15 s. Leucine–enkephalin was used as the lock mass in all analyses (in positive ion mode [M + H]^+^ = 556.2771) at a concentration of 0.5 μg/mL with a flow rate of 80 μL/min. The lock spray frequency was 20 s. Data were acquired in centroid mode. Analysis was carried out using full scan (MS scanning from *m*/*z* 100 to 1500).

### 3.5. Cell Culture and Treatment

PC12 cells were purchased from the cell resource center of the Chinese Academy of Medical Science (Beijing, China). The cells were stored in DMEM supplemented with 100 IU/mL penicillin, 100 μg/mL streptomycin, 10% fetal bovine serum, and 5% horse serum in humidified 5% CO_2_ atmosphere at 37 °C. In all experiments, cells in the exponential phase of growth were used.

For assessment of cytotoxicity induced by glutamate, PC12 cells were seeded in 96-well culture plates at a density of 105 cells/well. Different concentrations of glutamate (5, 10, 15, 30 mM) were incubated with PC12 cells for 12, 24 and 48 h respectively, then the cell viability was determined.

To research the cytotoxicity of a single compound on PC12 cells, cells were treated with compounds at the concentrations of 1.25, 2.5, 5 and 10 μM for 24 h. Then, the cell viability was determined.

To study the protective ability of the isolated compounds against toxicity induced by glutamate, PC12 cells were exposed to 15 mM glutamate for 24 h, then treated with the test compounds (1.25, 2.5, 5, 10 μM) for 24 h.

### 3.6. Measurement of Cell Viability

The cell viability was determined by the MTT test. In short, at the end of the indicated treatment, the PC12 cells were treated with MTT solution (final concentration of 0.5 mg/mL) for 4 h at 37 °C. Then, the formazan crystals formed in intact cells were solubilized with DMSO, and absorbance at 570 nm was measured with a microplate reader (Model 680, BIO-RAD Laboratories, Hercules, CA, USA). The cell viability is expressed as a percentage compared to the control group.

### 3.7. Annexin V/PI Double Staining

The Annexin V/PI assay kit was used to determine the cell apoptosis rate as per the manufacturer’s instruction.

The early stages of apoptosis were characterized by redistribution of phosphatidylserine to the external side of the cell membrane, which was due to the perturbations in the cellular membrane. This process provoked a flux of calcium, which was required by the Annexin V-labeled dye FITC to selectively bind to phosphatidylserine [32]. Thus, cells undergoing apoptosis were identified. Propidium iodide (PI) staining was used to distinguish early and late apoptotic cells from necrotic cells.

Cells were grown in 6-well plates until they reached a concentration of 2 × 10^5^ cells/mL. A negative control was prepared by incubating cells without glutamate. The cells after the incubation period were washed and collected in a cold environment, then recentrifuged. The supernatant was discarded and the cells were resuspended in 200 μL of Annexin-binding buffer. They were then incubated in the dark with both FITC-AnnexinV and PI for 15 min, and analyzed by a FACS Calibur flow cytometer (BD Biosciences, San Jose, CA, USA). All of the samples were detected in 1 h.

### 3.8. ROS, MDA and SOD Assays

The intracellular ROS level was identified by DCFH-DA (a nonfluorescent compound). DCFH-DA is enzymatically converted to the strongly fluorescent compound DCF in the existence of ROS. In short, PC12 cells were seeded onto a 6-well culture plate at a density of 6 × 10^5^ cells/well. After treating, the cells were washed by PBS and incubated with DCFH-DA at a final concentration of 10 μmol/L for 30 min at 37 °C in darkness. In order to remove the extracellular DCFH-DA, the cells were washed three times by PBS. Then, the fluorescence intensity of the DCF was measured with a fluorescent microplate reader at an excitation wavelength of 485 nm and an emission wavelength of 538 nm, the intracellular ROS levels were expressed as a percentage of controls.

MDA and SOD assays used the commercial kits (assay kits for determination of MDA and SOD) as per the manufacturer’s instructions. After the treatment, the PC12 cells were washed with cold PBS twice, harvested and centrifuged at 1000 *g* for 4 min, and then homogenized in 0.5 mL PBS. The homogenate was centrifuged at 4000 *g* for 15 min, and the supernatant was collected for MDA and SOD determination. The MDA content was determined using the thiobarbituric acid method, which formed a red compound with the maximum absorbance at 532 nm. The MDA content was calculated as follows:$$\mathrm{MDA}\;level=\;\frac{test\;tube\;absorbance-standard\;blank\;absorbance}{standard\;tube\;absorbance-blank\;tube\;absorbance}\;\times \frac{10\;\times dilution\;folds}{protein\;level}$$

The assay of total SOD relied on its ability to inhibit the oxidation of oxymine by the xanthine–xanthine oxidase system at 550 nm [33]. The calculation was performed as follows:
$$\mathrm{SOD}\;\mathrm{activity}=\;\frac{control\;tube\;absorbance-test\;tube\;absorbance}{control\;tube\;absorbance}\;÷50\%\;\times \frac{dilution\;folds}{protein\;level}$$

### 3.9. Statistical Analysis

The cell assays were carried out in triplicate. The results were expressed as mean values ± standard deviations. We analyzed the data using one-way ANOVA with Bonferroni’s correction. The *p*-values less than 0.05 were considered statistically significant.

## 4. Conclusions

A sensitive and efficient method employing UPLC-Q-TOF-MS was developed and 19 compounds were identified from MRB. Myricitrin and myricanol 11-sulfate were shown to be neuroprotective against the damage induced by glutamate on PC12 cells, and this neuroprotection was likely to prevent cell apoptosis through antioxidation. Our results not only suggest that MRB is a useful source of natural neuroprotection, but also provide a reliable basis to explore the potential of MRB as a food supplement in CNS diseases.

## Figures and Tables

**Figure 1 molecules-24-02993-f001:**
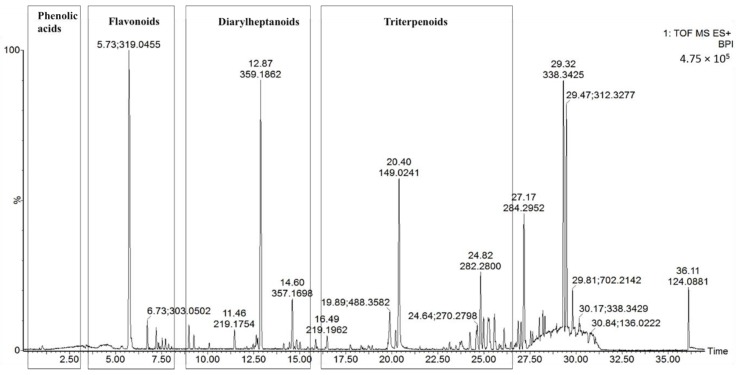
Representative based peak intensity (BPI) chromatograms of the MRB extract in the positive-ion mode.

**Figure 2 molecules-24-02993-f002:**
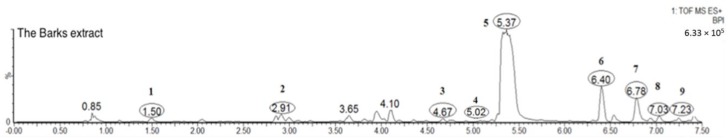
Typical total ion chromatogram of gallic acid (peak 1) and flavonoids (peaks 2–9) of the MRB extract.

**Figure 3 molecules-24-02993-f003:**
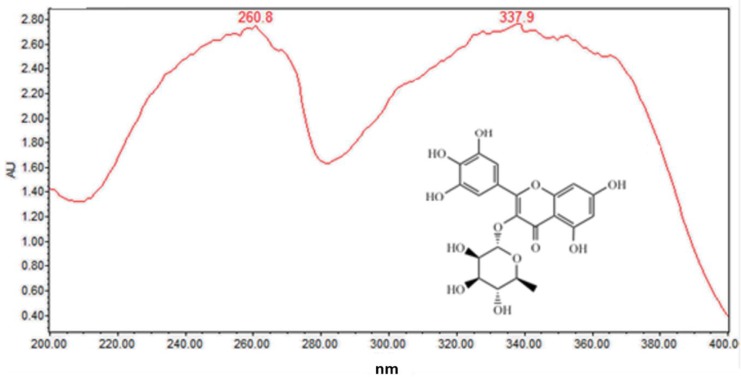
On-line UV-visible spectra of peak 5.

**Figure 4 molecules-24-02993-f004:**
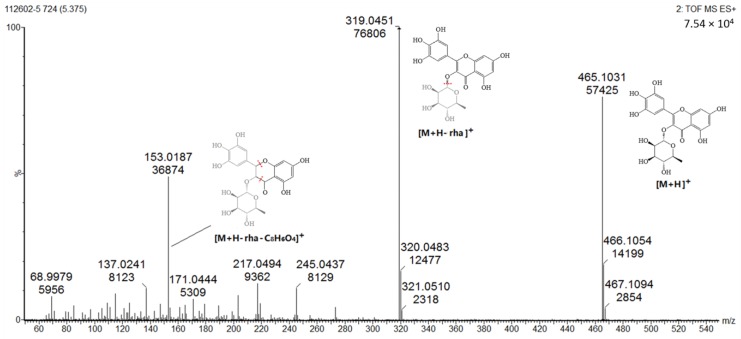
Product ion spectrum and proposed MS fragmentation mechanism of peak 5 in positive mode.

**Figure 5 molecules-24-02993-f005:**
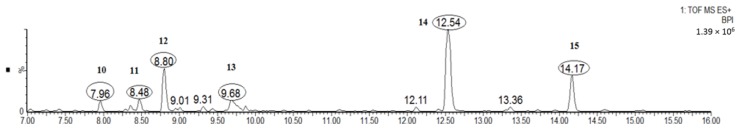
Typical total ion chromatogram of diarylheptanoids (peaks 10–15) of the MRB extract.

**Figure 6 molecules-24-02993-f006:**
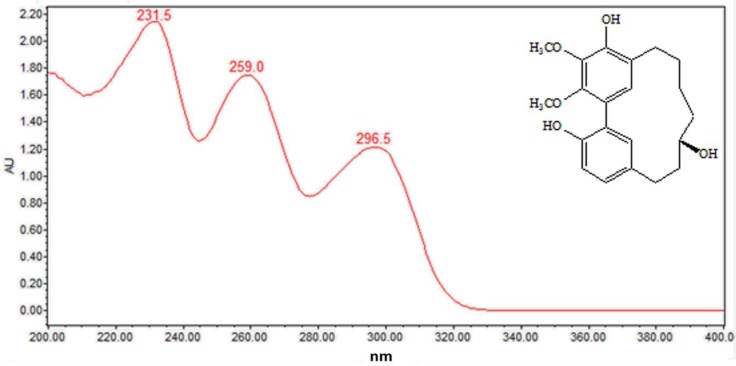
On-line UV–visible spectra of peak 14.

**Figure 7 molecules-24-02993-f007:**
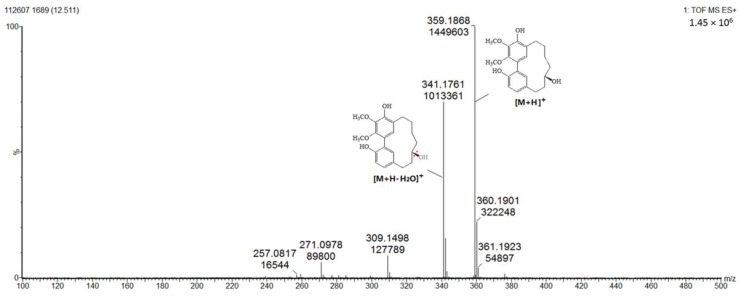
Product ion spectrum and proposed MS fragmentation mechanism of peak 14 in positive mode.

**Figure 8 molecules-24-02993-f008:**
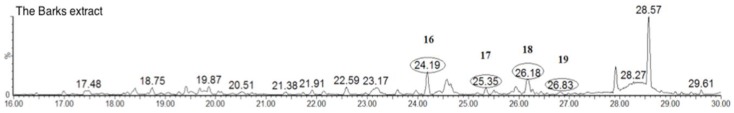
Typical total ion chromatogram of triterpenoids of the MRB extract.

**Figure 9 molecules-24-02993-f009:**
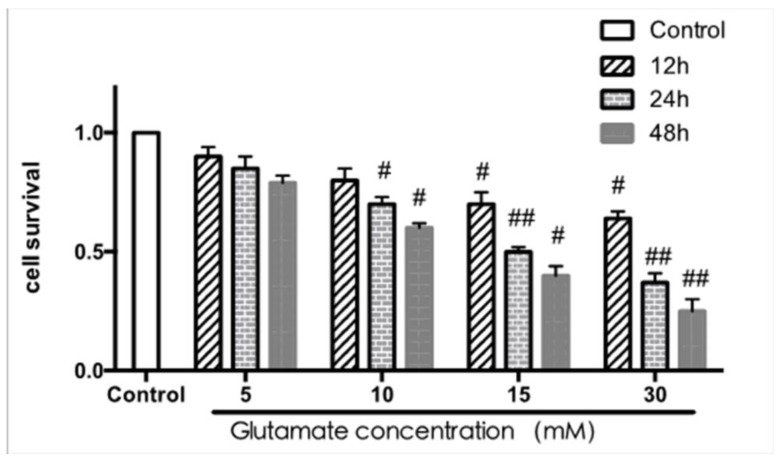
Effect of glutamate exposure on cell viability by MTT assay ($\bar{x}$ ± s*, n =* 4). ^#^
*p* < 0.05 ^##^
*p* < 0.01 vs. control group, same as below.

**Figure 10 molecules-24-02993-f010:**
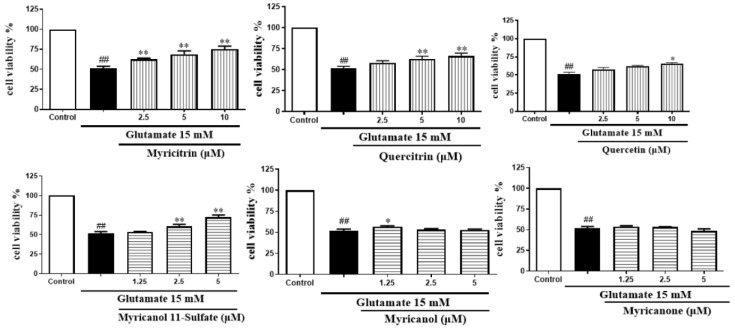
Effects of six major compounds on glutamate-induced PC12 cells through the MTT test ($\bar{x}$ ± s, *n* = 4) * *P* < 0.05 ** *P* < 0.01 vs. glutamate group; ^##^
*P* < 0.01 vs. control group.

**Figure 11 molecules-24-02993-f011:**
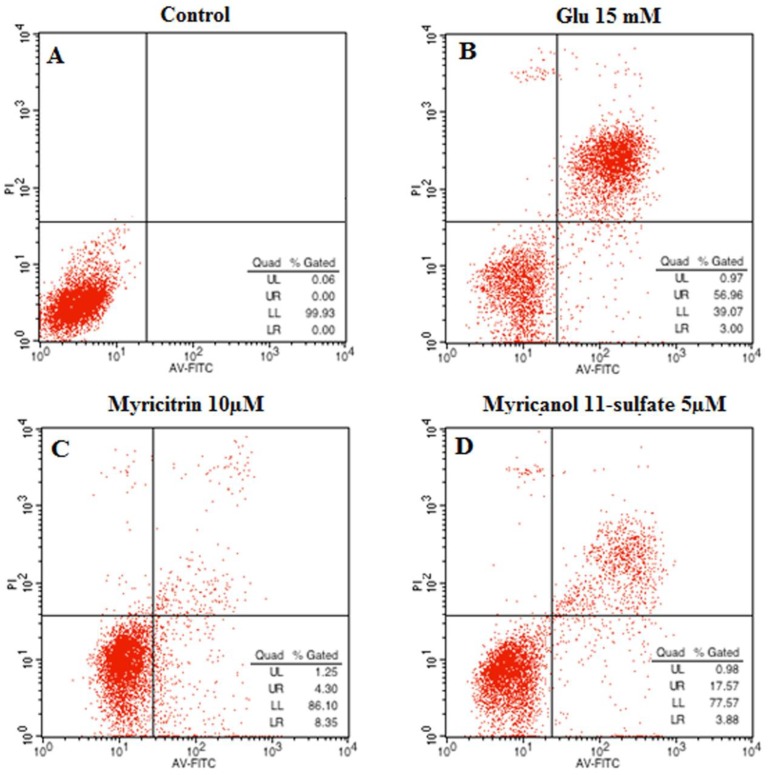
Effect of myricitrin and myricanol 11-sulfate on glutamate-induced apoptosis of PC12 cells analyzed by flow cytometer (**A**) control; (**B**) Cells with 15 mM glutamate treatment; (**C**) Cells treated with 10 μM myricitrin after glutamate-induced PC12 cell damage; (**D**) Cells treated with 5 μM myricanol 11-sulfate after glutamate-induced PC12 cell damage.

**Table 1 molecules-24-02993-t001:** Information of the 19 tentative identified compounds from the MRB extract.

Peak No.	t_R_ min	Molecular Formula	[M + H]^+^ *m/z*	MS/MS *m/z*	UV λnm	Tentative Identification
1	1.5	C_7_H_6_O_5_	171.1212	127.0708	219.3, 271.1	Gallic acid (std *)
2	2.9	C_27_H_30_O_16_	611.1321	303.0507, 153.0188	254.3, 353.8	Rutin (std *)
3	4.7	C_21_H_20_O_13_	481.3438	319.0451, 153.0187	260.2, 316.2	Myricetin hexoside
4	5.0	C_21_H_20_O_12_	465.0664	303.0526, 153.0208	261.7, 351.1	Quercetin hexoside
5	5.4	C_21_H_20_O_12_	465.1031	319.0451, 153.0187	260.4, 359.8	Myricitrin (std *)
6	6.4	C_21_H_20_O_11_	449.1089	303.0507, 229.0482, 153.0188	265.7, 338.1	Quercetin deoxyhexoside
7	6.8	C_15_H_10_O_8_	319.0455	153.0187	252.9, 372.7	Myricetin (std *)
8	7.0	C_15_H_10_O_7_	303.0502	153.0187	265.1, 365.9	Quercetin (std *)
9	7.2	C_15_H_10_O_6_	287.0938	153.0187	265.7, 338.1	Kaempferol
10	8.0	C_27_H_36_O_10_	521.2381	359.1859, 341.1768	216.2, 259.8, 305.1	Myricanol hexoside
11	8.5	C_27_H_36_O_10_	521.2381	359.1859, 341.1768	204.1, 259.6, 293.4	Myricanol hexoside
12	8.8	C_27_H_36_O_10_	521.2381	359.1859, 341.1768	222.4, 251.7, 294.7	Myricanol hexoside
13	9.7	C_21_H_26_O_8_S	439.1414	341.1750	213.2, 257.8, 295.3	Myricanol 11-sulfate (std *)
14	12.5	C_21_H_26_O_5_	359.1870	341.1761	231.5, 259.0, 296.5	Myricanol (std *)
15	14.2	C_21_H_24_O_5_	357.1840	339.1603, 325.1422	226.0, 258.4, 296.5	Myricanone (std *)
16	24.7	C_30_H_46_O_3_	455.3525			Uosolic Acid (std *)
17	25.5	C_30_H_50_O_2_	443.3889			Myricadoil (std *)
18	26.1	C_30_H_50_O_2_	443.3889			Uvaol (std *)
19	26.4	C_30_H_50_O	427.3943			Tarxerol (std *)

* The compounds were also identified by comparing the retention time with the authentic standards.

**Table 2 molecules-24-02993-t002:** Effects of Myricetrin and myricanol 11-sulfate on intracellular ROS, Malondialdehyde (MDA) and superoxide dismutase (SOD) levels in glutamate-induced damage to PC12 cells ($\bar{x}$ ± s, *n* = 6).

Group	Dose (μM)	ROS (% of Control)	MDA (nmol/mg Prot)	SOD (units/mg Prot)
Control	–	100 ± 1	0.76 ± 0.1	32 ± 4
Glutamate	15 mM	122 ± 6 ^##^	2.64 ± 0.3 ^##^	18 ± 1 ^##^
Myricanol 11-Sulfate	5	110 ± 3	0.8 ± 0.1 **	30 ± 1 **
Myricitrin	10	100 ± 6 **	0.78 ± 0.1 **	38 ± 2 **

** *p* < 0.01 versus glumate group, ^##^
*p* < 0.01 versus control group.

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
