# Peer review of "Study on the Material Basis of Neuroprotection of Myrica rubra Bark"

_molecules, 2019, doi:10.3390/molecules24162993_

Round 1

Reviewer 1 Report

The aim of the present study was to identify compounds from the Myrica rubra bark (MRB) extract and to investigate if 6 selected flavonoids and diarylheptanoids exert neuroprotective effects against glutamate-induced toxicity in PC12 cell.

In general, results of the study are potentially interesting and may be helpful in searching for natural compounds with beneficial effects on human health.

However, several issues can be attributed to this manuscript.

1. An important question is related to originality and novelty. The authors wrote (page 2, line 47): “Chemical studies reported MRB contains flavonoids, diarylheptanoids, triterpenes [7-11].”. If the composition of Myrica rubra bark has already been investigated/determined, the authors must clearly explain why this manuscript is important and worth publishing.

Besides, flavonoids and diarylheptanoids have already been recognized as promising neuroprotective agents ( for review see https://www.scipress.com/IJPPE.6.82.pdf)

2.The other important point of criticism is related to English language. The authors must seek professional assistance for English editing as parts of the manuscript are difficult to follow and there are many grammatical errors throughout the text.

Here are examples of some sentences that must be rewritten:

Page 2, lines 42-43: “Moreover, as the side effects of synthetic compounds, more and more attention has been paid to the search for natural plants.”

Page 2, lines 50-52: “The diarylheptanoids and phenolic compounds isolated from MRB were found to inhibit induction of NO synthase and overproduction of reactive oxygen species (ROS) [14,15], which is contribute to many degenerative nerve diseases.”

Page 2, lines 55-57: “In our searching for neuroprotective compounds from natural plants, we found MRB extract displayed better protection against glutamate-induced damage in PC12 cells (data not shown).” – unclear: better protection in comparison with what? Which data are not shown?

Page 4, lines 97-99: „Several well-resolved flavonoids chromatographic peaks observed from Figure 2. under the experimental conditions mentioned earlier, eight flavonoids (peaks 2-9) were provisionally identified.”

Page 7, lines 159-161: “The composition of triterpene composition is complex and exists structural isomers, most triterpenoids in the genus Myrica belongs to pentacyclic triterpenoids.”

Page 9, line2 220-224: “While the activity of SOD was significantly decreased from 32 ± 4 (the control value) to 18 ± 1 When treatment with 10 μM myricetrin and 5 μM myricanol 11-sulfate in the damaged cells induced by glutamate respectively, could significantly reduce intracellular ROS and MDA contents, as well as increase the SOD activity as compared with the control group.“

Furthermore, some sentences could be shortened to achieve the clarity.

E.g. Abstract, lines 18-21: “The results showed 19 compounds were identified, and myricitrin and myricanol 11-sulfate were proved to possess marked neuroprotection, which could prevent cell apoptosis through  alleviating oxidative stress by dropping out the levels of Reactive oxygen species and methane  dicarboxylic aldehyde, as well as enhancing the activities of superoxide dismutase.”

3.Page 1, lines 36-40: Molecular events that are described are characteristic for excitotoxicity. However, in PC12 cells treated with very high concentration of glutamate oxidative glutamate toxicity occurs (https://www.ncbi.nlm.nih.gov/pmc/articles/PMC4362409/). The authors must distinguish these two conditions and rewrite the paragraph.

4. The first paragraph (lines 32-35) is in the copy-paste manner adopted from ref. 1. Please re-write.

5.Values in the Figure 10 do not match with corresponding text.  It is not possible (from the graph) that viability after myricitrin and myricanol 11-sulfate was 75.15 ± 3.23 and 72.05 ± 2.09% respectively – please check

6. In comparison with other studies (some of them are indicated below), very small increase in ROS was observed in PC12 cells after exposure to 15 mM glutamate. Please comment.

https://www.researchgate.net/publication/283728977_Neuroprotective_Effects_of_Etidronate_and_233-Trisphosphonate_Against_Glutamate-Induced_Toxicity_in_PC12_Cells

https://www.ncbi.nlm.nih.gov/pubmed/10940457

https://www.ncbi.nlm.nih.gov/pmc/articles/PMC4685138/

http://citeseerx.ist.psu.edu/viewdoc/download?doi=10.1.1.320.9625&rep=rep1&type=pdf

7. The authors wrote that methane dicarboxylic aldehyde (MDA) content was determined using the thiobarbituric acid method. Was it methane dicarboxylic aldehyde or malondialdehyde? Usually the malondialdehyde is determined by thiobarbituric acid. Besides, based on the mechanism of oxidative glutamate toxicity it would be more relevant to monitor glutathione (GSH) content instead of monitoring SOD activity and MDA.

8. Regarding statistical analysis, the authors wrote that multiple group comparisons were performed using one-way analysis of variance (ANOVA) followed by Dunnett’s test. Dunnetts test is recommended when treated group are compared to control (it compares every mean to a control mean). After checking normality of the distribution, the results of this study have to be re-analysed with Tukey’s Test that compare every mean with every other mean.

9. Regarding the neuroprotective potential of myriceirin and myricanol 11-sulfate, please comment their ability to penetrate across the blood brain barrier.

Minor issues

Conclusions in the Abstract (lines 22-23) could be written more precisely. The authors wrote that “ Myrica rubra barks are a useful natural source of neuroprotection and have the potential for development drug of central nervous system diseases.”

“Several active compounds from Myrica rubra barks may offer neuroprotection and have the potential for the development of new drugs against central nervous system diseases associated with excess levels of glutamate. “

Lines 103-105 and 109-110: duplication, almost identical text, please delete/rewrite

Line 102 almost identical to line 111 (peak 5), please delete/rewrite

Please mark peak 4 in Figure 2.

The numbers in the text (line 103, absorption maxima) do not match with red numbers in Figure 3. Please check/explain for those readers that are not familiar with the method.

Again, the numbers in the text (line 136, absorption maxima) do not match with red numbers in Figure 6. Please check/explain.

For peak 14 in Table 1 there is a number m/z 341.1768, whereas in the text (line 140) there is a 341.1768 – please check/explain. Similarly, for peak 13 in Table 1 there is 341.175, whereas in the text (line 141) it is 341.1768.

Page 6, lines 157-159: “According to the literature and chemical standard obtained in our previous phytochemical studies confirmation, peak 16, 17, 18 and 19 were identified as uosolic acid, myricadoil, uvaol and tarxerol, respectively.” – please add references

Page 8 - change myricetrin to myricitrin in Figure 10

Pages 7, lines 179-184: This whole paragraph is not written clearly. The authors must explain how, i.e. on what basis, they selected six major compounds and name these compounds. It must be clearly written which 3 compounds did not affect viability and which three diarylheptanoids were slightly cytotoxic.

line 244 – JECFA – explain abbreviation

line 352 - please name the commercial kits

Reviewer 2 Report

1. Please give information about PC12 cell line when first used in the paper, for eg. neuroblastic cells.

2. Flow diagram suggest that myricitrin at 10um concerntartion significantly reduce total apoptotic cells. however, myricitrin also increase early apoptotic cells as compared to glu 15mm treated cells. What does another think about the mechanism that migration offer to protect the cells from apoptosis.

3. Since the authors used isolated compounds for the cells treatment after glutamate treatment, the compound showed therapeutic effects. it would be suggestive of pre-treating cells with the compounds and then treat with glutamate to study their protective effects.

Round 2

Reviewer 1 Report

The manuscript can be accepted for publication.